# A Recipe for Causal Graph Regression: Confounding Effects Revisited

**Yujia Yin** [* 1] **Tianyi Qu** [* 2 3] **Zihao Wang** [4] **Yifan Chen** [1]

## Abstract

Through recognizing causal subgraphs, causal graph learning (CGL) has risen to be a promising approach for improving the generalizability of graph neural networks under out-of-distribution (OOD) scenarios. However, the empirical successes of CGL techniques are mostly exemplified in classification settings, while regression tasks, a more challenging setting in graph learning, are overlooked. We thus devote this work to tackling causal graph regression (CGR); to this end we reshape the processing of confounding effects in existing CGL studies, which mainly deal with classification. Specifically, we reflect on the predictive power of confounders in graph-level regression, and generalize classification-specific causal intervention techniques to regression through a lens of contrastive learning. Extensive experiments on graph OOD benchmarks validate the efficacy of our proposals for CGR. The model implementation and the code are provided on https://github.com/causal-graph/CGR.

## 1. Introduction

Causal graph learning (CGL) (Lin et al., 2021) holds particular importance due to its relevance in fields such as drug discovery (Qiao et al., 2024) and climate modeling (Zhao et al., 2024). However, previous CGL studies focus on classification settings. Some of them cannot be directly extended to regression tasks, such as property prediction (Rollins et al., 2024), traffic flow forecasting (Li et al., 2021), and credit risk scoring (Ma et al., 2024), because the transition from finite to infinite support makes discrete labels unavailable. Graphs thus cannot be informatively grouped. A systematical understanding of how CGL techniques should be adapted to graph-level regression is still under-explored.

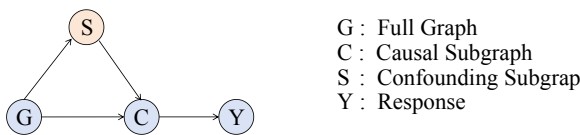

G : Full Graph
C : Causal Subgraph
S : Confounding Subgraph
Y : Response

*Figure 1.* Structural causal model (SCM) for graph regression.

The core methodology of causal learning involves the identification and differentiation of causal features from confounding ones. As shown in Figure 1, causal features $C$ are those directly deciding responses $Y$, whereas confounding features $S$ (shorthand for "spurious") solely present spurious correlations. Therefore, understanding how causal features (as well as confounding features) and responses interact plays a central role in practical designing of causal learning methods. From this perspective, causal graph regression (CGR) warrants specialized handling since the interaction between features and responses therein is significantly different from classification. Furthermore, regression is in general a more challenging task than classification, and techniques working for classification, Perceptron (Rosenblatt, 1958) for example, may not apply to regression.

Specifically in CGL, the identification of causal subgraphs is seemingly transferable since this step, explicitly or implicitly, relies on the calculation of mutual information and is compatible with both settings (c.f. Section 3.2). However, the empirical performance of this vanilla adaptation on regression tasks is dwarfed by empirical risk minimization (Vapnik, 1991, ERM) w.r.t. least squares loss (see the results in Sections 5.3 and 5.4).

To crack CGR, we revisit the processing of confounding effects, which conceptually constitutes causal graph learning along with causal subgraph identification as shown in Figure 1. Existing CGL methods, such as CAL (Sui et al., 2022) and DisC (Fan et al., 2022), are built on a strong assumption that confounding subgraphs contain strictly no predictive power. We reflect on this assumption and speculate it is hardly practical due to the contradiction with real-world observations: in molecular property prediction, for example, molecular weight is noncausal to toxicity while does exhibit strong correlations.

---

*Equal contribution [1]Hong Kong Baptist University [2]SF Tech [3]Zhejiang University [4]Hong Kong University of Science and Technology. Correspondence to: Tianyi Qu <qutianyi@sf-express.com>, Yifan Chen <yifanc@hkbu.edu.hk>.

*Proceedings of the 42nd International Conference on Machine Learning*, Vancouver, Canada. PMLR 267, 2025. Copyright 2025 by the author(s).

In this work, we develop an enhanced graph information bottleneck (GIB) loss function, which no longer takes the strong assumption. Moreover, some confounding effect processing techniques, such as backdoor adjustment (Sui et al., 2022; 2024) and counterfactual reasoning (Guo et al., 2025), heavily rely on discrete label information and cannot be adapted to regression at all. We follow the principle of those methods and generalize it from class separation to instance discrimination; the discrimination principle aligns with the philosophy of contrastive learning (CL) and CL techniques are therefore leveraged to tackle CGR in our proposal.

Following the intuition, we develop a new framework for causal graph regression, which spotlights the confounding effects within. In summary, our contributions are as follows:

- To the best of our knowledge, we are the first to explicitly consider the predictive role of confounding features in graph regression tasks, a critical yet overlooked aspect in graph OOD generalization.

- We introduce a new causal intervention approach that generates random graph representations by leveraging a contrastive learning loss to enhance causal representation, outperforming label-dependent methods.

- Extensive experiments on OOD benchmarks demonstrate that our method significantly improves generalization in graph regression tasks.

## 2. Related Work

Out-of-distribution (OOD) challenges in graph learning has drawn significant attention, particularly in methods aiming to disentangle causal and confounding factors (Ma, 2024). Existing approaches can be broadly categorized into invariant learning (Wu et al., 2022a), causal modeling (Sui et al., 2024), and stable learning (Li et al., 2022).

**Invariant learning** focuses on identifying features that remain stable across different environments, filtering out spurious correlations in the process. While not explicitly grounded in causal reasoning, prior studies (Wang & Veitch, 2022; Mitrovic et al., 2020) have highlighted its inherent connection to causality. Methods in invariant learning, such as CIGA (Chen et al., 2022), GSAT (Miao et al., 2022), and GALA (Chen et al., 2024), aim to learn invariant representations by isolating causal components.

However, these approaches are typically designed for classification tasks, limiting their out-of-distribution (OOD) generalization capability in regression settings. Post-hoc methods, such as PGExplainer (Luo et al., 2020) and Reg-Explainer (Zhang et al., 2023), attempt to discover invariant subgraphs after training. However, these methods fail to equip the model with the ability to learn invariant representa-tations during the training process.

**Causal modeling** leverages structural causal models (SCMs) to improve the performance of graph neural networks (GNNs) on out-of-distribution (OOD) data. These approaches incorporate various traditional causal inference techniques, such as backdoor adjustment (e.g., CAL (Sui et al., 2022), CAL+ (Sui et al., 2024)), frontdoor adjustment (e.g., DSE (Wu et al., 2022c)), instrumental variables (e.g., RCGRL (Gao et al., 2023)), and counterfactual reasoning (e.g., DisC (Fan et al., 2022)). By simulating causal inter-ventions through supervised training, these methods aim to achieve OOD generalization. However, they often disregard the predictive potential of confounding features, which hin-ders effective disentanglement. Moreover, the supervised loss functions tailored for classification tasks are not easily adaptable to regression problems, as the inherent complexity of regression introduces additional challenges.

**Stable learning** aims to ensure consistent performance across environments by reweighting samples or balancing covariate distributions. For example, StableGNN (Fan et al., 2023) employs a regularizer to reduce the influence of confounding variables. However, such methods often rely on heuristic reweighting strategies, which may not fully disentangle causal from confounding factors.

In addition to graph-based approaches, traditional machine learning methods have also explored causality in regression tasks. For instance, Pleiss et al. (2019) observed that causal features tend to concentrate in a low-dimensional subspace, whereas non-causal features are more randomly distributed. Similarly, Amini et al. (2020) proposed a framework for learning continuous targets by placing an evidence prior on a Gaussian likelihood function and training a non-Bayesian neural network to infer the hyperparameters of the evidence distribution. These methods highlight the potential of lever-aging causal insights for improved regression performance.

## 3. Preliminaries and Notations

Along this paper, we denote a graph $G$ as $(\boldsymbol{A}, \boldsymbol{X})$. Here, $\boldsymbol{A} \in \{0, 1\}^{n \times n}$ is the adjacency matrix indicating connec-tivity among $n$ nodes ($\boldsymbol{A}_{ij} = 1$ if nodes $i$ and $j$ are con-nected, otherwise 0); $\boldsymbol{X} \in \mathbb{R}^{n \times d}$ is the node feature matrix, where each row $\boldsymbol{X}_i$ represents the $d$-dimensional feature vector of node $i$. The regression task in graph learning is to learn a function $f : G \mapsto y$, where $y \in \mathbb{R}$ denotes the response for the graph $G$.

### 3.1. Causal Graph Learning

In causal graph learning, a graph $G$ can be split into a **causal subgraph** $C$ and a **confounding subgraph** $S$. This process is non-trivial and our proposed paradigm will hinge on the output of this process. We follow the definition in Sui et al.

(2022) and first introduce the construction of the causal subgraph $C$:

$$C := (\boldsymbol{M}_{\text{edge}} \odot \boldsymbol{A}, \boldsymbol{M}_{\text{node}} \cdot \mathbf{X}), \qquad (1)$$

where the mask matrix $\boldsymbol{M}_{\text{edge}} \in [0,1]^{n \times n}$ and the diagonal matrix $\boldsymbol{M}_{\text{node}}$ (whose diagonal elements are in $[0,1]$) will filter out the non-causal nodes and edges. The confounding subgraph is then the "complement": $S := G - C$.

In our framework, these masks $\boldsymbol{M}_{\text{edge}}$ and $\boldsymbol{M}_{\text{node}}$ are not pre-defined. Instead, they are learnable soft masks, generated by MLPs conditioned on the representations of $G$. The parameters of these MLPs are optimized end-to-end as part of the overall model training, enabling the model to autonomously learn how to construct $C$ and $S$. Further architectural details are provided in Appendix B and illustrated in Figure 2.

Notably, mutual information plays an essential role in CGL, and we introduce its calculation, exemplified by the mutual information between the hidden embedding vectors (learned by a graph neural network) of the causal subgraphs and the original graphs, as follows:

$$I(C;G) := \mathbb{E}_{C,G}\left[\log p(C \mid G)/p(C)\right], \qquad (2)$$

where we follow the convention in CGL literature and abuse the notation $C, G$ to represent a random variable following the **underlying distribution of embedding pairs** $H_{g,i}$'s and $H_{c,i}$'s. In particular, those hidden embeddings are assumed Gaussian and the joint distribution can thus be well-estimated by sample embedding pairs. We refer readers interested to Miao et al. (2022, Appendix A) for more details. Moreover, the computation/approximation of the mutual information terms is a crucial component in causal graph learning, while still under-explored for CGR; we will dissect the computation of our proposed terms in Section 4.2 through deriving the variational bounds.

### 3.2. Graph Information Bottleneck

The information bottleneck (Tishby et al., 2000; Tishby & Zaslavsky, 2015, IB) principle aims to balance the trade-off between preserving the information necessary for prediction and discarding irrelevant redundancy. Specifically, IB suggests to maximize $I(Z;Y)$ while minimizing $I(Z;X)$ for regular data compression, where $Z$ is the compressed representation, $X$ is the input, and $Y$ is the response.

Graph information bottleneck (GIB) (Wu et al., 2020) extends the IB principle to graph-structured data, facilitating the identification of subgraphs that are most relevant for predicting graph-level responses. By minimizing the mutual information $I(C;G)$ between the extracted causal subgraph $C$ and the original graph $G$, GIB reduces redundant information. However, GIB alone does not guarantee the extraction of a purely causal subgraph, as isolating causal effects re-

quires additional interventions (Miao et al., 2022; Chen et al., 2022).

Formally, the GIB objective is expressed as:

$$-I(C;Y) + \alpha I(C;G), \qquad (3)$$

where $I(C;Y)$ quantifies the predictive information retained by $C$ (and thus needs to maximize). $I(C;G)$ serves as a regularizer to exclude irrelevant details from the original graph; the parameter $\alpha$ controls the trade-off between information preservation and compression.

### 3.3. Causal Intervention in GNNs

We borrow the structural causal model (SCM) diagram in Figure 1 to illustrate the causal intervention techniques. As shown in Figure 1, the graph $G$ decides both the causal subgraph $C$ and the confounding subgraph $S$, and the former $C$ affects the prediction of response $Y$. In more detail,

- $C \leftarrow G \rightarrow S$: Graph data $G$ encodes both $C$, which directly impacts $Y$, and $S$, which introduces spurious correlations.
- $S \rightarrow C \rightarrow Y$: The causal feature $C$ has the potential to predict $Y$ not only directly but also indirectly through its influence along this backdoor path $S \rightarrow C \rightarrow Y$.

In causal inference, confounder $S$ incurs spurious correlations, preventing the discovery of underlying causality. To address this issue, backdoor adjustment methods focus on the interventional effect $P(Y|\text{do}(C))$, and suggest to estimate it by stratifying over $S$ and calculating the conditional distribution $P(Y|C,S)$ (Pearl, 2014; Sui et al., 2024).

## 4. Revisiting Confounding Effects for CGR

In this section, we present a causal graph regression paradigm that integrates an enhanced graph information bottleneck (GIB) objective with causal discovery, reshaping the processing of confounding effects in CGL.

### 4.1. Overview

We first provide an overview of how graph inputs are turned into regression outputs. As shown in Figure 2, we follow the framework of Sui et al. (2024) and first encode graph embeddings $H_{g,i}$'s using a GNN-based encoder. Attention modules are then adopted to generate soft masks for extracting causal and confounding subgraphs (c.f. Equation (1)). These subgraphs are processed through two GNN modules ($\mathcal{G}_c$ and $\mathcal{G}_s$) with shared parameters to extract causal ($H_{c,i}$'s) and confounding ($H_{s,i}$'s) representations, which are passed through distinct readout layers for regression.

The optimization features an enhanced graph information bottleneck (GIB) loss $L_{\text{GIB}}$, comprising the causal part $L_c$ and the confounding part $L_s$, to disentangle causal signals

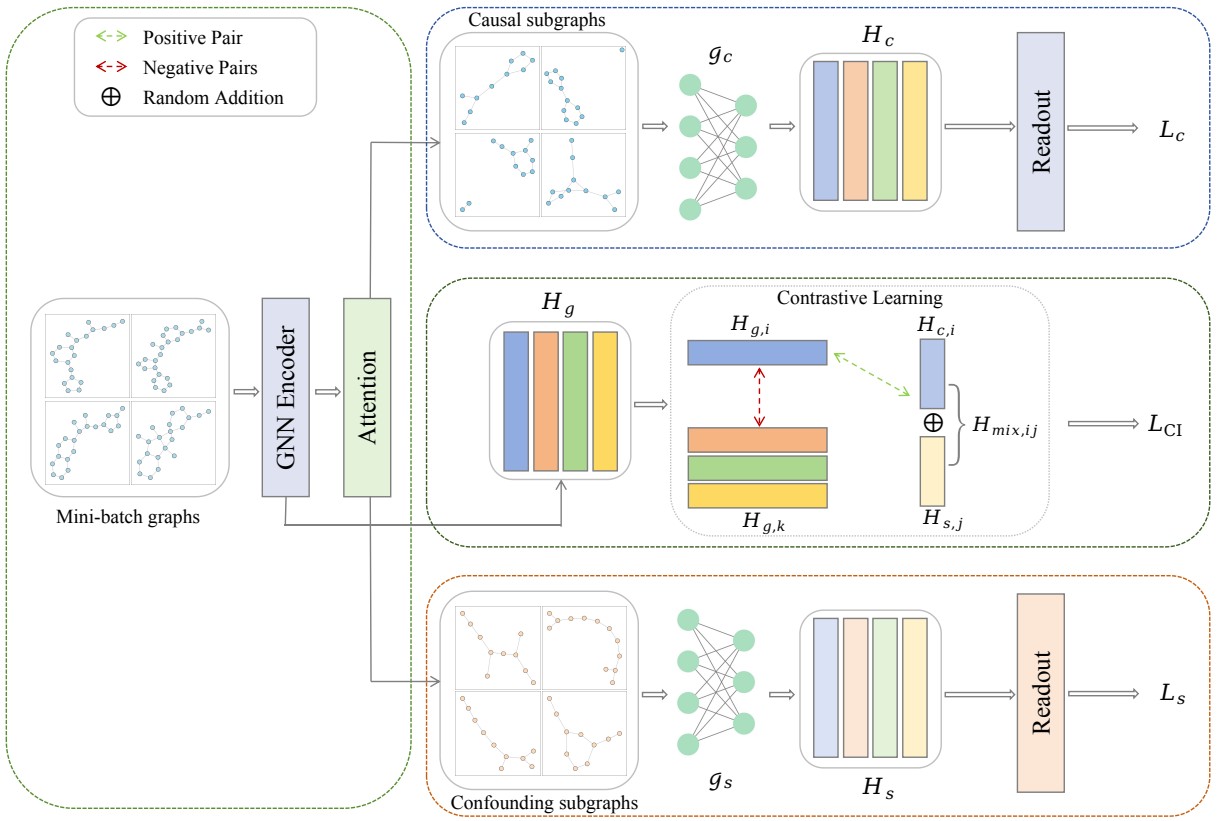

*Figure 2.* Given a mini-batch of graphs, (1) the GNN encoder computes the graph embeddings $H_g$, and an attention layer generates soft masks to extract causal and confounding subgraphs. (2) GNN $\mathcal{G}_c$ processes the causal subgraph $C$, generates its representation $H_c$, and employs readout to predict responses; it is optimized with causal subgraph loss $L_c$. (3) GNN $\mathcal{G}_s$, sharing parameters with $\mathcal{G}_c$, processes the confounding subgraph $S$, generates $H_s$, and applies readout for prediction; it is optimized with confounding subgraph loss $L_s$. (4) For causal intervention, contrastive learning guides the process. Given a graph $H_{g,i}$, the positive sample is a mixed graph $H_{\mathrm{mix},ij}$ from random addition, while any other graph $H_{g,k}$ serves as the negative sample. The causal intervention loss $L_{\mathrm{CI}}$ is used accordingly.

(c.f. Section 4.2). Also, counterfactual samples ($H_{\mathrm{mix},ij}$) are generated by randomly injecting confounding representations into causal ones; unsupervised learning is then performed, guided by contrastive-learning-based causal intervention loss $L_{\mathrm{CI}}$ (c.f. Section 4.3). More implementation details of the overall framework are deferred to Appendix B.

### 4.2. Enhanced GIB Objective

CGL adopts the GIB objective to extract subgraphs that retain essential predictive information while excluding redundant components (Zhang et al., 2023), which aligns with the disentanglement of causal subgraph $C$ and confounding subgraph $S$ in CGL. Original GIB assumes the confounding subgraph $S$ is pure noise and cannot predict the response $Y$ (Chen et al., 2022), while as we discussed in Section 1 $S$ may still contain information that is predictive of the re-

sponse $Y$. In its current form, the GIB framework overlooks this aspect, causing the model to allocate all $Y$-relevant information to $C$ and to potentially lose meaningful content.

This limitation leads to incomplete causal disentanglement, which impacts the generalization of models to out-of-distribution (OOD) settings. To overcome this issue, we propose an enhanced GIB loss function that takes the predictive roles of both $C$ and $S$ into consideration. By introducing mutual information terms on $S$ during optimization, we avoid overburdening $C$ with all relevant information, and consequently enable a more precise disentanglement.

Overall, our enhanced GIB objective is defined as follows:

$$-I(C;Y) + \alpha I(C;G) - \beta I(S;Y), \qquad (4)$$

which formally extends the original GIB objective by introducing a confounder-related term $I(S;Y)$ to capture the

predictive capacity of $S$, along with a parameter $\beta$. In particular, we intentionally exclude the $I(S;G)$ term because, in the SCM diagram of Figure 1, $S$ primarily introduces shortcut rather than directly encoding causality; overly imposing structural regularization on $S$ can disrupt disentanglement and lead to suboptimal separation between $C$ and $S$. Notably, the conceptual objective (4) is incomputable in practice. We devote the remainder of this subsection to the practical computation of Equation (4) for CGR.

**Variational bounds for approximating $I(C;G)$.** The mutual information $I(C;G)$ is mathematically defined based on the marginal distributin $p(C) = \sum_G p(C|G)p(G)$. Since $p(C)$ is intractable, a variational distribution $q(C)$ is introduced and induces an upper bound:

$$I(C;G) \leq \mathbb{E}_{p(G)}\big[\mathrm{KL}\big(p(C \mid G)\|q(C)\big)\big]. \qquad (5)$$

To efficiently compute the KL divergence in Equation (5), we follow the literature (Chechik et al., 2003; Kingma et al., 2013) and assume that $p(C \mid G)$ and $q(C)$ are multivariate Gaussian distributions:

$$p(C \mid G) = \mathcal{N}(\mu_\phi(G), \Sigma_\phi(G)), \quad q(C) = \mathcal{N}(0, I), \quad (6)$$

where $\mu_\phi(G)$ and $\Sigma_\phi(G)$ are the mean vector and covariance matrix estimated by GNNs. To simplify computation and stabilize training, we further assume $\Sigma_\phi(G)$ is an identity matrix, removing the need to learn covariance parameters. This simplification is not only practical but also theoretically justified, as any full-rank covariance can be whitened without loss of generality (Chechik et al., 2003, Appendix A). $\mathrm{KL}\big(p(C \mid G)\|q(C)\big)$ then reduces to:

$$
\begin{aligned}
&\frac{1}{2}\Big[\mathrm{tr}(\Sigma_\phi(G)) + \|\mu_\phi(G)\|^2 - d - \log\det\Sigma_\phi(G)\Big] \\
&= \frac{1}{2}\|\mu_\phi(G)\|^2.
\end{aligned}
\qquad (7)
$$

where $d$ is the dimensionality of $C$. Further substituting Equation (7) into Equation (5), we obtain an upper bound for $I(C;G)$:

$$I(C;G) \leq \frac{1}{2}\mathbb{E}_{p(G)}\big[\|\mu_\phi(G)\|^2\big], \qquad (8)$$

which serves as an easy-to-compute proxy for $I(C;G)$.

**Variational bounds for approximating $I(C;Y), I(S;Y)$.** We first recall $I(C;Y)$ mathematically reads:

$$I(C;Y) = H(Y) - H(Y \mid C), \qquad (9)$$

where $H(Y)$ denotes the entropy of $Y$, representing the overall uncertainty in the target variable. Since $H(Y)$ remains constant, maximizing $I(C;Y)$ reduces to minimizing the conditional entropy $H(Y \mid C)$, given by:

$$H(Y \mid C) = -\mathbb{E}_{C,Y}[\log p(Y \mid C)]. \qquad (10)$$

The computation of $H(Y \mid C)$ is supposed to hinge on the hidden embeddings $H_{c,i}$'s produced by a GNN $\mathcal{G}_c$ (see Section 4.1); we model the conditional distribution $p(Y \mid H_c)$ as a Gaussian distribution:

$$p(Y \mid H_c) = \mathcal{N}(Y; \mu_{(c)}, \sigma_{(c)}^2), \qquad (11)$$

where $\mu_{(c)}$ and $\sigma_{(c)}^2$ represent the scalar conditional mean and variance of $Y$ (estimated by networks) given a causal subgraph representation $H_c$. The probability density function for this Gaussian is:

$$p(Y \mid H_c) = \frac{1}{\sqrt{2\pi\sigma_{(c)}^2}} \exp\left(-\frac{(Y - \mu_{(c)})^2}{2\sigma_{(c)}^2}\right). \qquad (12)$$

Substituting Equation (12) into Equation (10), we can further approximate $H(Y \mid C)$ through empirical data:

$$\frac{1}{N}\sum_{i=1}^{N}\left[\frac{(Y_i - \mu_{(c),i})^2}{2\sigma_{(c),i}^2} + \frac{1}{2}\log(2\pi\sigma_{(c),i}^2)\right], \qquad (13)$$

where $N$ represents sample size, $Y_i$ is the target response for the $i$-th sample. and $\mu_{(c),i}$ and $\sigma_{(c),i}^2$ are the corresponding mean and variance of $Y$ given $H_{c,i}$.

If a constant conditional variance (i.e., $\sigma_{(c)}^2 = 1$) is assumed, a choice adopted for stability and aligning with approaches in (Nix & Weigend, 1994; Yu et al., 2024), then $I(C;Y)$ (or, equivalently, $-H(Y \mid C)$) reduces to the least squares loss:

$$
\begin{aligned}
&-\frac{1}{N}\sum_{i=1}^{N}\left[\frac{(Y_i - \mu_{(c),i})^2}{2\sigma_{(c),i}^2} + \frac{1}{2}\log(2\pi\sigma_{(c),i}^2)\right] \\
&\propto -\frac{1}{N}\sum_{i=1}^{N}(Y_i - \mu_{(c),i})^2,
\end{aligned}
\qquad (14)
$$

which turns to the **causal subgraph objective** $L_{\mathrm{CP}}$.

Similarly, the mutual information $I(S;Y)$ can induce the **confounding subgraph objective**

$$L_{\mathrm{SP}} \propto -\frac{1}{N}\sum_{i=1}^{N}(Y_i - \mu_{(s),i})^2. \qquad (15)$$

Empirically, we employ two independent readout layers to compute the causal and confounding subgraph mean $\mu_{(c),i}$'s and $\mu_{(s),i}$'s.

In summary, our enhanced GIB objective can be decomposed into two distinct loss components: the causal subgraph loss $L_c(G, C, Y) = -I(C;Y) + \alpha I(C;G)$ and the confounding subgraph loss $L_s(S, Y) = -I(S;Y)$. The complete enhanced GIB objective we propose is:

$$
\begin{aligned}
L_{\mathrm{GIB}} &= L_c + \beta L_s \\
&= -I(C;Y) + \alpha I(C;G) - \beta I(S;Y),
\end{aligned}
$$

and in practice we use $-L_{\mathrm{CP}} + \alpha\mathbb{E}_{p(G)}\big[\|\mu_\phi(G)\|^2\big] - \beta L_{\mathrm{SP}}$.

### 4.3. Causal Intervention

To further strengthen causal learning in CGR, we introduce a causal intervention loss and reshape the processing of confounding effects therein. In general, our approach injects randomness at the graph level by randomly pairing confounding subgraphs with target causal subgraphs from the entire dataset. By generating counterfactual graph representations through the random combination of these subgraphs, we effectively implement causal intervention.

This strategy can be understood as an implicit realization of backdoor adjustment (Pearl, 2014) in the representation space. In existing research on graph classification tasks (Fan et al., 2022; Sui et al., 2024), causal intervention is typically modeled by predicting $P(Y|C, S)$ through intervened graphs, adjusting for causal effects by comparing predictive distributions under different confounding conditions. However, in regression tasks, $Y$ is a continuous variable, and directly modeling $P(Y|C, S)$ becomes significantly more challenging. To overcome this, we follow the spirit of contrastive learning to get rid of the reliance on explicit labels.

In more detail, following Sui et al. (2022), we use a random addition method to pair the confounding subgraph with the target causal subgraph , which gives $H_{\mathrm{mix}}$:

$$H_{\mathrm{mix},ij} = H_{c,i} + H_{s,j}. \tag{16}$$

Comparing the predictions of $H_{\mathrm{mix}}$ with the original graph's labels, as shown in Sui et al. (2022), can inadvertently force the mixed graph to discard all confounding effects, thereby nullifying the intended causal disentanglement.

To mitigate this issue, we suggest learning causal representations through contrastive learning. Specifically, the causal subgraph, when combined with different confounding subgraphs, consistently produces mixed graph representations that are aligned with the original graph representation. This formulation enables the model to learn causal subgraphs that are invariant across varying confounders, and to avoid the causal subgraphs boiled down to non-informative ones.

To achieve this, we propose a causal intervention loss guided by contrastive learning. Specifically, the method aligns the representation of the original graph with that of its corresponding random mixture graph, while simultaneously ensuring that representations of unrelated graphs remain distinct. In implementation, draw inspiration from the InfoNCE loss (Oord et al., 2018), we treat $H_g$ and $H_{\mathrm{mix}}$ from the same causal subgraph as positive pairs, and $H_g$ with representations of other graphs within the batch as negative pairs. Formally, the mixed graph contrastive loss is defined as:

$$L_{\mathrm{CI}} = -\frac{1}{B} \sum_{i=1}^{B} \log \frac{\exp(\mathrm{sim}(H_{g,i}, H_{\mathrm{mix},ij}))}{\sum_{k=1, k \neq i}^{B} \exp(\mathrm{sim}(H_{g,i}, H_{g,k}))}, \tag{17}$$

where $B$ is the batch size, $H_{\mathrm{mix},ij}$ is the representation of the mixed graph combining the $i$-th causal subgraph and the $j$-th confounding subgraph, and $H_{g,i}$ is the representation of the original graph.

*Remark* 4.1. The ultimate loss used in our paradigm is a simple combination of the GIB objective and the causal intervention loss: $L = L_{\mathrm{GIB}} + \lambda L_{\mathrm{CI}}$.

## 5. Experiments

In this section, we evaluate the prediction performance and OOD generalization ability of our method. We comprehensively compare our method with existing models to demonstrate the superior generalization ability of our method on regression tasks. We briefly introduce the dataset, baselines, and experimental settings here.

### 5.1. Datsets

**GOOD-ZINC.** GOOD-ZINC is a regression task in the GOOD benchmark (Gui et al., 2022), which aims to test the out-of-distribution performance of real-world molecular property regression datasets from the ZINC database (Gómez-Bombarelli et al., 2018). The input is a molecular graph containing up to 38 heavy atoms, and the task is to predict the restricted solubility of the molecule (Jin et al., 2018; Kusner et al., 2017). GOOD-ZINC includes four specific OOD types: Scaffold-Covariate, Scaffold-Concept, Size-Covariate, and Size-Concept. Scaffold OOD involves changes in molecular structures, while Size OOD varies graph size. Each can manifest as Covariate Shift ($P(X)$ changes, $P(Y|X)$ remains stable) or Concept Shift (spurious correlations in training break in testing).

**ReactionOOD-SOOD.** In addition to the GOOD benchmark, we also used three S-OOD datasets in the ReactionOOD benchmark (Wang et al., 2023), namely Cycloaddition (Stuyver et al., 2023), E2&$S_N$2 (von Rudorff et al., 2020), and RDB7 (Spiekermann et al., 2022), which are designed to extract information outside the structural distribution during molecular reactions. Cycloaddition and RDB7 have two domains: Total Atom Number (where the total number of atoms in a reaction exceeds the training range) and First Reactant Scaffold (where the first reactant has a new molecular scaffold unseen in training), while E2&$S_N$2 dataset contains reactions with molecules whose scaffold cannot be properly defined, which prevents the scaffold from being an applicable domain index for this dataset. The definitions of two shifts Covariate and Concept in ReactionOOD are consistent with those in GOOD.

### 5.2. Baselines and Setup

As our framework is general and aims to address distribution shifts, we compare it against several baseline methods.

*Table 1.* OOD generalization performance on GOOD-ZINC dataset, with **boldface** being the best and underline being the runner-up.

| GOOD-ZINC | SCAFFOLD | | | | SIZE | | | |
| --- | --- | --- | --- | --- | --- | --- | --- | --- |
| | COVARIATE | | CONCEPT | | COVARIATE | | CONCEPT | |
| | ID | OOD | ID | OOD | ID | OOD | ID | OOD |
| ERM | 0.1188±0.0030 | 0.1660±0.0093 | 0.1174±0.0013 | 0.1248±0.0018 | 0.1222±0.0061 | 0.2331±0.0169 | 0.1304±0.0010 | 0.1406±0.0002 |
| IRM | 0.1258±0.0033 | 0.2313±0.0243 | 0.1176±0.0052 | 0.1245±0.0062 | 0.1217±0.0014 | 0.5840±0.0039 | 0.1331±0.0045 | 0.1338±0.0011 |
| VREx | 0.0978±0.0016 | 0.1561±0.0021 | 0.1928±0.0021 | 0.1271±0.0020 | 0.1841±0.0009 | 0.2276±0.0005 | 0.1206±0.0008 | 0.1289±0.0039 |
| Mixup | 0.1348±0.0025 | 0.2157±0.0098 | 0.1192±0.0026 | 0.1296±0.0049 | 0.1431±0.0070 | 0.2573±0.0042 | 0.1625±0.0121 | 0.1660±0.0063 |
| DANN | 0.1152±0.0021 | 0.1734±0.0005 | 0.1284±0.0031 | 0.1289±0.0020 | 0.1053±0.0081 | 0.2254±0.0140 | 0.1227±0.0008 | 0.1271±0.0039 |
| Coral | 0.1252±0.0043 | 0.1734±0.0034 | 0.1173±0.0029 | 0.1260±0.0024 | 0.1164±0.0004 | 0.2243±0.0147 | 0.1246±0.0062 | 0.1270±0.0020 |
| CIGA | 0.1568±0.0034 | 0.2986±0.0041 | 0.1926±0.0120 | 0.2415±0.0115 | 0.1500±0.0001 | 0.6102±0.0148 | 0.3560±0.0160 | 0.3240±0.0451 |
| DIR | 0.2483±0.0056 | 0.3650±0.0032 | 0.2510±0.0001 | 0.2619±0.0076 | 0.2515±0.0529 | 0.4224±0.0679 | 0.4831±0.0823 | 0.3630±0.0872 |
| GSAT | 0.0890±0.0031 | 0.1419±0.0043 | 0.0928±0.0029 | 0.0999±0.0029 | 0.0876±0.0032 | 0.2112±0.0033 | 0.1002±0.0013 | 0.1043±0.0001 |
| Ours | **0.0514±0.0061** | **0.1046±0.0007** | **0.0659±0.0041** | **0.0518±0.0007** | **0.0466±0.0034** | **0.1484±0.0033** | **0.0577±0.0008** | **0.0580±0.0004** |

Empirical Risk Minimization (ERM) (Vapnik, 1991) serves as a non-OOD baseline for comparison with OOD methods. We consider both Euclidean and graph-based state-of-the-art OOD approaches: (1) Euclidean OOD methods include IRM (Arjovsky et al., 2019), VREx (Krueger et al., 2021), GroupDRO (Sagawa et al., 2019), DANN (Ganin et al., 2016), Coral (Sun & Saenko, 2016), and Mixup (Zhang, 2017); (2) Graph OOD methods include CIGA (Chen et al., 2022), GSAT (Miao et al., 2022), and DIR (Wu et al., 2022b).

For a fair comparison, all methods are implemented with consistent architectures and hyperparameters, ensuring that performance differences arise solely from the method itself. To provide reliable results, each experiment is repeated three times with different random seeds, and we report the mean and standard error of the results. Detailed settings and hyperparameter configurations are described in Appendix A.4.

### 5.3. Results of GOOD

As shown in Table 1, our proposed method achieves SOTA performance on GOOD-ZINC, consistently outperforming all baseline methods across both domains (Scaffold and Size) and under different distribution shifts (Covariate and Concept). Specifically, in terms of Mean Absolute Error (MAE), our method demonstrates significant improvements in both in-distribution (ID) and out-of-distribution (OOD) settings.

For instance, in the Scaffold domain under the Covariate shift, our method achieves an MAE of 0.0514±0.0061 (ID) and 0.1046±0.0007 (OOD), outperforming GSAT, the next-best method, by 42.2% in ID and 26.3% in OOD performance. Similarly, under the Concept shift, our method achieves 0.0659±0.0041 (ID) and 0.0518±0.0007 (OOD), representing improvements of 29.0% and 48.1%, respectively, over GSAT.

In the Size domain, our method also achieves remarkable results. Under the Covariate shift, it achieves an MAE of 0.0466±0.0034 (ID) and 0.1484±0.0033 (OOD), which translate to 46.8% lower ID error and 29.7% lower OOD

error compared to GSAT. Similarly, under the Concept shift, our approach yields an MAE of 0.0577±0.0008 (ID) and 0.0580±0.0004 (OOD), improving upon GSAT by 42.4% and 44.4%, respectively.

In addition to achieving lower MAE values, our method exhibits significantly reduced variances compared to other approaches, highlighting its stability under diverse conditions. These findings confirm the strong generalization capability of our method across different domains and types of distributional shifts.

### 5.4. Results of ReactionOOD

Table 2 and Table 3 highlight the robust generalization ability of our method across multiple datasets and evaluation settings, as measured by RMSE. Our method achieves the best OOD performance in 6 out of 10 cases and ranks second in 2 cases. Notably, in cases where another method outperforms ours, the performance gap is within a small margin.

For instance, in the Cycloaddition dataset, under the total atom number domain with a concept shift, Our method achieves an OOD RMSE of 5.53 ± 0.12, outperforming all baseline methods. While some non-causal baselines (e.g., Coral in this specific setting, achieving an ID RMSE of 4.10 ± 0.05 versus our 4.41 ± 0.22) might get better ID performance by exploiting spurious but predictive features, such approaches can become less reliable under OOD conditions (e.g., Coral's OOD RMSE degrades to 5.74 ± 0.04). In contrast, our method's focus on identifying and removing these spurious features contributes to its stable and superior OOD performance. Even in other Cycloaddition cases where ours ranks second, such as the same domain with a covariate shift, the OOD RMSE (4.42 ± 0.24) is only 0.06 away from the best-performing method (4.36 ± 0.15).

In RDB7, a smaller dataset within the ReactionOOD where causal inference can be more difficult, our method achieves the lowest OOD RMSE (15.73 ± 0.37) under the concept shift. Our method's principled focus on true causal features, which leads to better OOD generalization ability and stabil-

*Table 2.* OOD generalization performance on Cycloaddition and RDB7 dataset.

| DATASET | METHODS | FIRST REACTANT SCAFFOLD | | | | TOTAL ATOM NUMBER | | | |
| | | COVARIATE | | CONCEPT | | COVARIATE | | CONCEPT | |
| | | ID | OOD | ID | OOD | ID | OOD | ID | OOD |
|---|---|---|---|---|---|---|---|---|---|
| CYCLOADDITION | ERM | 4.38±0.04 | 4.80±0.38 | 4.79±0.03 | 5.60±0.02 | **3.77±0.01** | **4.36±0.15** | 4.22±0.04 | 5.69±0.03 |
| | IRM | 15.30±0.05 | 21.16±0.01 | 17.55±0.03 | 18.64±0.25 | 17.53±0.17 | 17.44±0.14 | 23.14±0.02 | 22.56±0.01 |
| | VREX | 5.54±0.02 | 6.69±0.48 | 5.02±0.05 | 6.14±0.09 | 4.79±0.03 | 5.22±0.06 | 4.92±0.14 | 6.39±0.04 |
| | MIXUP | 4.51±0.04 | 5.24±0.83 | 4.90±0.01 | 5.90±0.05 | 3.90±0.13 | 4.53±0.03 | 4.11±0.09 | 5.93±0.13 |
| | DANN | 4.42±0.03 | 4.68±0.12 | 4.81±0.01 | 5.75±0.06 | 3.87±0.05 | 4.65±0.10 | 4.18±0.02 | 5.68±0.10 |
| | CORAL | **4.36±0.07** | 4.95±0.30 | 4.82±0.03 | 5.72±0.16 | 4.39±0.59 | 5.05±0.48 | **4.10±0.05** | 5.74±0.04 |
| | CIGA | 5.26±0.04 | 5.67±0.04 | 5.30±0.29 | 5.64±0.03 | 4.93±0.05 | 6.62±1.09 | 5.03±0.09 | 6.21±0.06 |
| | DIR | 4.94±0.02 | 5.31±0.79 | 5.85±0.20 | 6.30±0.38 | 5.52±0.03 | 6.86±0.05 | 5.21±0.12 | 7.09±0.03 |
| | GSAT | 4.42±0.05 | 4.63±0.05 | 4.87±0.01 | 5.69±0.01 | 3.81±0.01 | 4.56±0.01 | 4.12±0.04 | 5.64±0.11 |
| | OURS | 4.57±0.13 | **4.22±0.09** | **4.53±0.04** | **5.37±0.05** | 4.06±0.01 | 4.42±0.24 | 4.41±0.22 | **5.53±0.12** |
| RDB7 | ERM | 10.28±0.05 | 22.95±0.90 | 11.38±0.08 | **14.81±0.05** | 10.86±0.01 | 7.66±0.55 | **11.28±0.15** | 15.79±0.24 |
| | IRM | 59.87±0.02 | 76.51±0.46 | 65.72±0.13 | 63.03±0.13 | 63.55±0.02 | 69.06±0.37 | 81.14±0.02 | 46.84±0.42 |
| | VREX | 16.62±0.18 | **21.89±0.02** | 14.62±0.04 | 18.28±0.09 | 14.60±0.01 | 13.84±0.07 | 34.66±1.56 | 32.59±3.28 |
| | MIXUP | 10.76±0.07 | 23.49±0.09 | 11.89±0.05 | 15.64±0.10 | 11.13±0.02 | 10.78±0.17 | 11.66±0.04 | 17.21±0.28 |
| | DANN | 10.28±0.05 | 23.54±0.07 | 11.28±0.01 | 14.93±0.05 | 10.77±0.22 | 8.29±0.10 | 11.34±0.05 | 16.28±0.15 |
| | CORAL | 10.30±0.12 | 22.19±0.63 | **11.12±0.03** | 14.81±0.06 | 10.61±0.01 | 8.04±0.14 | 11.33±0.08 | 16.13±0.08 |
| | CIGA | 14.97±0.75 | 30.08±0.84 | 18.68±1.94 | 21.35±1.34 | 16.48±0.69 | 19.12±1.85 | 20.58±1.54 | 18.53±1.30 |
| | DIR | 14.34±0.68 | 26.99±0.49 | 17.13±1.76 | 20.18±1.86 | 14.03±2.06 | 15.01±0.98 | 13.52±0.51 | 16.60±1.09 |
| | GSAT | 10.52±0.04 | 23.45±0.11 | 11.26±0.25 | 14.85±0.12 | 10.80±0.01 | 8.66±0.10 | 11.58±0.03 | 16.08±0.41 |
| | OURS | **10.12±0.08** | 23.11±0.46 | 11.26±0.02 | 14.94±0.25 | **10.51±0.08** | **6.84±0.32** | 11.46±0.06 | **15.73±0.37** |

*Table 3.* OOD generalization performance on E2&S$_N$2 dataset.

| METHODS | COVARIATE | | CONCEPT | |
| | ID | OOD | ID | OOD |
|---|---|---|---|---|
| ERM | 4.45±0.04 | 5.47±0.27 | 4.87±0.02 | 5.04±0.02 |
| IRM | 11.61±0.18 | 21.54±1.07 | 20.95±0.02 | 17.57±0.03 |
| VREX | 4.58±0.02 | 5.48±0.13 | 10.75±1.54 | 8.77±2.31 |
| MIXUP | 4.55±0.09 | 5.55±0.01 | 4.69±0.08 | 5.11±0.01 |
| DANN | 4.51±0.06 | 5.38±0.04 | **4.48±0.10** | 5.04±0.02 |
| CORAL | 4.44±0.11 | 5.68±0.20 | 4.54±0.02 | **4.97±0.07** |
| CIGA | 5.05±0.35 | 6.57±0.52 | 4.65±0.26 | 5.39±0.47 |
| DIR | 5.61±0.26 | 6.59±0.31 | 6.56±0.34 | 6.29±0.11 |
| GSAT | 4.55±0.01 | 5.69±0.05 | 4.55±0.09 | 5.04±0.03 |
| OURS | **4.40±0.03** | **4.83±0.10** | 4.53±0.12 | 5.03±0.09 |

ity. Even though causal methods generally face challenges in smaller datasets (Guo et al., 2020), our approach consistently outperforms other listed causal intervention baselines such as CIGA in all RDB7 settings. In the E2&S$_N$2 dataset, our method delivers the best OOD RMSE (4.83 ± 0.10) under the covariate shift and achieves highly competitive results under the concept shift (5.03 ± 0.09).

As noted in OOD-GNN (Tajwar et al., 2021), no method consistently performs best on every dataset due to varying distribution shifts and inductive biases. Our approach, designed under more general and weaker assumptions which do not assume that spurious features are non-predictive, aims to tackle a wider range of real-world distribution shifts.

### 5.5. Effectiveness of OURS in Classification Task

To validate the generality and effectiveness of our proposed losses, $L_{GIB}$ and $L_{CI}$, we conduct ablation studies on the GOOD-Motif dataset under the size domain setting. The results, evaluated in terms of accuracy, are reported on the OOD dataset, as shown in Figure 3. The ablation study on $L_{GIB}$ aims to examine our hypothesis that confounders possess certain predictive power; thus, this experiment excludes the causal intervention loss $L_{CI}$ Conversely, the ablation study on $L_{CI}$ evaluates whether the contrastive learning-driven causal intervention loss can independently achieve strong OOD performance. Therefore, in this experiment, we do not incorporate the predictive power of confounding factors.

**Predictive power of confounding subgraphs.** The left panel compares minimizing confounding subgraph prediction alone versus introducing constraints to model their predictive ability. The results show that ignore the predictive role of confounding subgraphs leads to incomplete disentanglement and weaker OOD generalization, demonstrating that accounting for their influence is crucial.

**Effectiveness of contrastive learning.** The right panel compares using predictions from randomly generated counterfactual graphs as causal intervention loss versus our proposed contrastive learning loss. The results show that our contrastive learning approach, initially validated in regression tasks, is equally effective in classification tasks, highlighting its general applicability.

These studies confirm the importance of explicitly modeling confounding subgraphs and the robustness of our contrastive learning loss for OOD generalization. More experimental results are provided in the Appendix A.5.

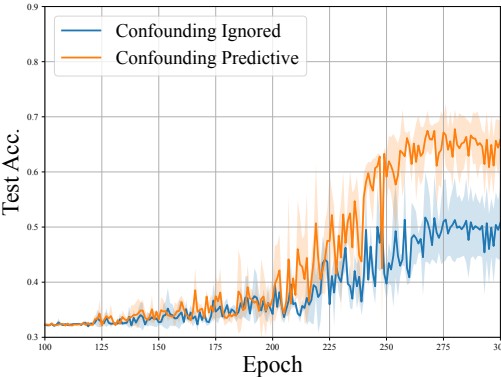 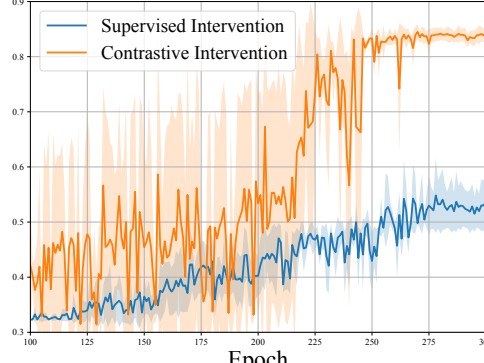

*Figure 3.* Ablation study on confounder predictive power (left) and causal intervention methods (right) for OOD generalization on GOOD-Motif.

# 6. Conclusion

In this work, we propose a recipe for causal graph regression through reshaping the processing of confounding effects in existing CGL classification-specific techniques. In particular, we develop an enhanced graph information bottleneck (GIB) loss function which highlights the impact of confounding effects and consequently benefits the recognition of causal subgraphs. Moreover, we revisit the causal intervention technique, which randomly combines causal subgraphs and confounder from the same class (label) to eliminate confounding effects. Adapting this technique to regression requires removal of label information; to this end, we analyze the principle of causal intervention and propose to connect it with unsupervised contrastive learning loss. Experimental results on graph OOD benchmarks demonstrate the effectiveness of our proposed techniques in improving the generalizability of graph regression models.

## Acknowledgements

We sincerely thank the Area Chair and the anonymous reviewers for their valuable feedback and constructive suggestions, which helped improve this work. The authors acknowledge funding from Research Grants Council (RGC) under grant `22303424` and GuangDong Basic and Applied Basic Research Foundation under grant `2025A1515010259`.

## Impact Statement

This paper presents work whose goal is to advance the field of Machine Learning. There are many potential societal consequences of our work, none which we feel must be specifically highlighted here.

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

# A. Supplementary Experiments

## A.1. GOOD Benchmark

The Graph Out-Of-Distribution (GOOD) benchmark is the most comprehensive and authoritative benchmark for assessing the OOD generalization of graph learning models. It includes 11 datasets, covering six graph-level and five node-level tasks, with 51 dataset splits across covariate shift, concept shift, and no shift scenarios. Among them, nine datasets focus on classification (binary and multi-class), one (GOOD-ZINC) on regression, and one (GOOD-PCBA) on multi-objective binary classification. GOOD is the first benchmark to incorporate both covariate and concept shifts within the same domain, enabling controlled comparisons. It evaluates 10 state-of-the-art OOD methods, including four tailored for graphs, resulting in 510 dataset-model combinations. As a result, GOOD provides a systematic and rigorous framework for benchmarking OOD generalization in graph learning

## A.2. ReactionOOD Benchmark

The ReactionOOD benchmark is a specialized out-of-distribution (OOD) evaluation framework designed to systematically assess the generalization capabilities of machine learning models in predicting the kinetic properties of chemical reactions. It introduces three distinct levels of OOD shifts—structural, conditional, and mechanistic—and comprises six datasets, all formulated as regression tasks. Structural OOD (S-OOD) examines variations in reactant structures, including shifts based on total atomic count (E2 & SN2) and reactant scaffolds (RDB7, Cycloaddition). Conditional OOD (C-OOD) investigates the effect of environmental conditions on kinetic properties, considering shifts in temperature (RMG Lib. T) and combined temperature-pressure settings (RMG Lib. TP). Mechanistic OOD (M-OOD) explores the impact of different reaction mechanisms (RMG Family) on kinetic property predictions.

## A.3. GOOD-ZINC Dataset Details

Table 4 presents the number of graphs/nodes in different dataset splits for the GOOD-ZINC dataset. The dataset is analyzed under three types of distribution shifts: covariate, concept, and no shift. Each row represents the number of graphs/nodes in training, in-distribution (ID) validation, ID test, out-of-distribution (OOD) validation, and OOD test sets. The no-shift scenario serves as a baseline with no distributional difference between training and test sets.

*Table 4.* Details of GOOD-ZINC dataset.

| Dataset | Shift | Train | ID validation | ID test | OOD validation | OOD test |
|---|---|---|---|---|---|---|
| | covariate | 149674 | 24945 | 24945 | 24945 | 24946 |
| GOOD-ZINC | concept | 101867 | 21828 | 21828 | 43539 | 60393 |
| | no shift | 149673 | 49891 | 49891 | - | - |

## A.4. Experimental Settings

We use the GOOD-ZINC dataset from the GOOD benchmark and the S-OOD tasks from ReactionOOD, excluding other OOD tasks from ReactionOOD as they are still under maintenance. Our baseline results on ReactionOOD have been acknowledged by the original authors. We use a three-layer GIN as the backbone model, with 300 hidden dimensions, which is consistently applied in both OURS and baseline models. The model is trained for 300 epochs, with the learning rate adjusted using the cosine annealing strategy. The initial learning rate is set to 0.001, with a minimum value of 1e-8 . For the OURS model, all tunable hyperparameters in the loss function $L$ are set to 0.5.

## A.5. Ablation Studies

**Effectiveness Analysis**   To evaluate the effectiveness of the proposed loss functions $L_{GIB}$ and $L_{CI}$ in improving the model's OOD generalization ability, we conducted a series of ablation studies across four ood datasets: ZINC, Cycloaddition, E2SN2, and RDB7. Ours w/o BO serves as the baseline model, where both loss functions are removed, and only the causal subgraph readout layer's $l_1$ loss is used for optimization. Ours w/o GIB ablates $L_{GIB}$, eliminating the constraint on confounding subgraphs to assess the impact of removing confounder control on generalization. Conversely, Ours w/o CI removes $L_{CI}$ while keeping $L_{GIB}$, allowing us to examine the contribution of the causal intervention loss to OOD generalization. Ours represents the complete model, incorporating both loss functions for optimization. Notably, ZINC is

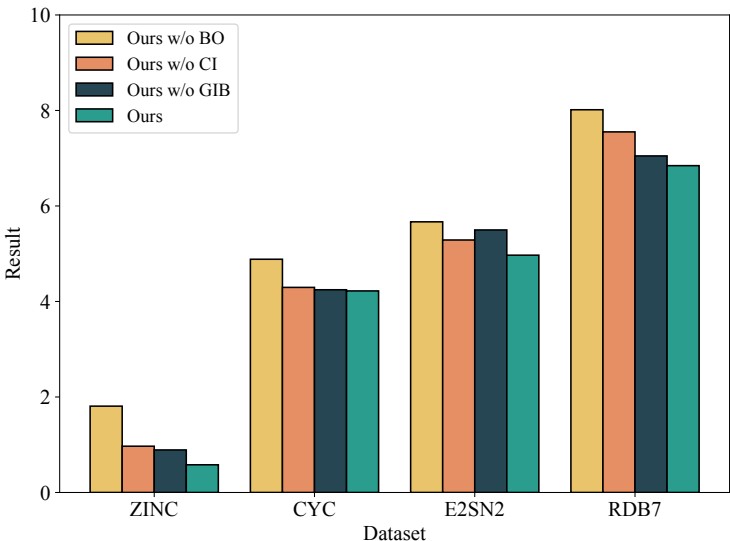

*Figure 4.* The comparison of different components.

evaluated using MAE, while the other datasets adopt RMSE as the evaluation metric. Given that the ZINC results are small (approximately 0.0x), we scale them by a factor of 10 in the Figure 4 for better visualization and comparison.

The results reveal several key insights. The full model (green) consistently achieves the lowest RMSE across all datasets, demonstrating the effectiveness of jointly applying both the enhanced GIB loss and the CI loss. Removing both components (yellow) leads to the worst performance, confirming that both components are essential. Between the two losses, removing CI (orange) generally causes a larger degradation than removing GIB (blue), suggesting that CI plays a more dominant role. On E2SN2, however, GIB contributes more significantly. These results indicate that GIB and CI provide complementary benefits, and that using both yields the best OOD generalization.

**Parameter Sensitivity Analysis** In this experiment, we analyzed the sensitivity of loss function hyperparameters under different settings in the Cycloaddition dataset, focusing on two key components of our proposed loss function: the hyperparameter $\lambda$ for the causal intervention term and $\alpha$, $\beta$ for the confounding constraint term. The results in Figure 5

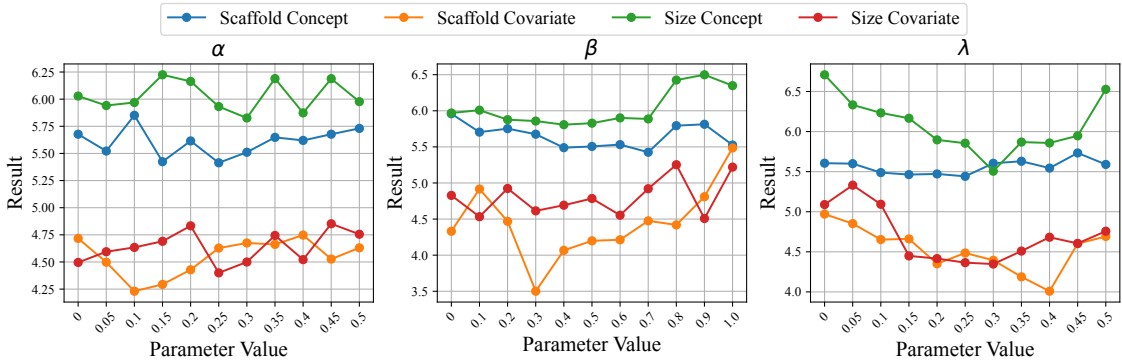

*Figure 5.* Parameter sensitivity.

indicate that, there is no clear trend toward getting better or worse for $\alpha$. For $\beta$, which balances the GIB loss, there is a gradual increase in RMSE when it is too large, especially in scaffold-covariate settings, suggesting an optimal range around 0.3–0.6. For $\lambda$, which controls the causal intervention loss, has the strongest impact. A suitable parameter interval (0.2–0.4) consistently leads to lower RMSE, while overly large or small $\lambda$ causes performance degradation, especially in the size-concept setting. This demonstrates the importance of carefully tuning $\lambda$ to achieve effective OOD generalization.

## B. Framework Details

Given a GNN-based encoder $f(\cdot)$ and a graph $G_i = (A_i, X_i)$, the graph representation is computed as:

$$H_{g,i} = f(A_i, X_i), \tag{18}$$

Then, to estimate attention scores, inspired by (Sui et al., 2022), we utilize separate MLPs for nodes and edges. The node attention scores, which can be seen as the node-level soft mask can be computed as:

$$\mathrm{M_{node}}, \bar{\mathrm{M}}_{\mathrm{node}} = \sigma(\mathrm{MLP_{node}}(H_{g,i})), \tag{19}$$

where $\sigma$ denotes the softmax operation applied across attention dimensions. Similarly, edge-level soft masks are determined by concatenating node embeddings from connected edges, followed by an edge-specific MLP:

$$\mathrm{M_{edge}}, \bar{\mathrm{M}}_{\mathrm{edge}} = \sigma(\mathrm{MLP_{edge}}([H_{g,i}[\mathrm{row}], H_{g,i}[\mathrm{col}]])), \tag{20}$$

These soft masks serve as weighting mechanisms, allowing the model to focus on the most relevant nodes and edges while maintaining differentiability.

Next, we decompose the initial graph to causal and confounding attened-subgraph:

$$\mathrm{C}_i = \left\{ A_i \odot M_{edge}, X_i \odot M_{node} \right\}, \tag{21}$$

$$\mathrm{S}_i = \left\{ A_i \odot \bar{M}_{edge}, X_i \odot \bar{M}_{node} \right\}. \tag{22}$$

To encode these subgraphs, $C_i$ and $S_i$ are processed through a pair of GNNs with shared parameters, extracting causal and confounding representations $H_c$ and $H_s$, respectively. Finally, the representations of the two subgraphs are respectively used to obtain the predictions of the regression task through the corresponding readout layers.

## C. Variational Bounds for the GIB Objective

The mutual information $I(C; G)$ quantifies the dependency between $C$ and $G$ and is defined as:

$$I(C; G) = \mathbb{E}_{p(C,G)} \left[ \log \frac{p(C \mid G)}{p(C)} \right]. \tag{23}$$

However, computing the marginal distribution $p(C) = \sum_G p(C \mid G)p(G)$ is intractable, to overcome this challenge, we approximate $p(C)$ with a variational distribution $q(C)$. Substituting $q(C)$ into Eq. (23), we reformulate $I(C; G)$ as:

$$I(C; G) = \mathbb{E}_{p(C,G)} \left[ \log \frac{p(C \mid G)}{q(C)} \right] - \mathrm{KL}\big(p(C)\|q(C)\big). \tag{24}$$

The KL divergence term $\mathrm{KL}\big(p(C)\|q(C)\big)$ is non-negative, providing an upper bound for $I(C; G)$:

$$I(C; G) \leq \mathbb{E}_{p(G)}\big[\mathrm{KL}\big(p(C \mid G)\|q(C)\big)\big]. \tag{25}$$

