# OpenReview forum: "A Recipe for Causal Graph Regression: Confounding Effects Revisited"
_ICML.cc/2025/Conference — ICML 2025 poster_

### Official Review · Reviewer_iAdk · 2025-03-07

**Overall Recommendation:** 3

**Summary:**

This paper addresses the challenge of adapting causal graph learning (CGL) techniques from classification to regression tasks, introducing a framework called causal graph regression (CGR). The authors identify two key innovations: (1) an enhanced graph information bottleneck loss function that, unlike previous approaches, acknowledges the predictive power of confounding features rather than treating them as pure noise, and (2) a contrastive learning approach for causal intervention that can operate without discrete class labels. By combining these techniques, their method generates counterfactual graphs through random combinations of causal and confounding subgraphs, enabling the model to learn invariant representations. Extensive experiments on GOOD-ZINC and ReactionOOD benchmarks demonstrate performance improvements in out-of-distribution settings.

**Claims And Evidence:**

Claim 1: Existing causal graph learning can not work well for graph regression tasks.

Support: 1. Citations. 2. Experimental results in Section 5. This claim is supported with evidence.

Claim 2: The reason that existing CGL methods don't work well for regression tasks stems from an assumption that confounding subgraphs contain strictly no predictive power.

Support: I don't find clear supports for this claim. I personally deduce that this is related to the optimization dynamics related to tasks (classification and regression).

**Essential References Not Discussed:**

[1] is closely related but not discussed at all.

[1] Wu, Tailin, et al. "Graph information bottleneck." Advances in Neural Information Processing Systems 33 (2020): 20437-20448.

**Experimental Designs Or Analyses:**

For evaluation, the authors use appropriate benchmark datasets:
GOOD-ZINC: A standard graph regression dataset explicitly designed for OOD testing
ReactionOOD: Specialized datasets (Cycloaddition, E2&SN2, RDB7) with regression tasks for chemical reactions
The evaluation criteria are reasonable.

**Methods And Evaluation Criteria:**

Methodologically, the paper extends causal graph learning to regression tasks by:
Rethinking how confounding effects should be handled in regression (acknowledging predictive power of confounding features rather than treating them as pure noise)
Developing a contrastive learning approach to replace label-dependent causal intervention techniques that don't transfer well to regression.
These two designs make sense, while the motivation for the second approach is not clearly stated. Authors thoroughly introduce the modeling of $I(C;Y)$ and $I(C;G)$. I wonder if there's any difference from those introduced in existing papers. If not, authors should focus on two mechanisms introduced in this paper.

For evaluation, the authors use appropriate benchmark datasets:
GOOD-ZINC: A standard graph regression dataset explicitly designed for OOD testing
ReactionOOD: Specialized datasets (Cycloaddition, E2&SN2, RDB7) with regression tasks for chemical reactions
The evaluation criteria are reasonable.

**Other Comments Or Suggestions:**

Definition in Section 3: You are considering a set of graphs but not a single graph.
5.2. baselines and Setup -> Baselines

**Other Strengths And Weaknesses:**

I think this paper suffers from two limitations. First, the scope is excessively narrow, focusing on a highly specific problem within graph regression. Second, method design lacks clear motivation and support, as the authors fail to adequately justify their methodological choices or connect them to established theoretical principles.

**Questions For Authors:**

I notice that the proposed methods achieve much larger gains on GOOD datasets than ReactionOOD datasets. Do you have any ideas on the reason?

**Relation To Broader Scientific Literature:**

* Out-of-Distribution Generalization for Graphs: This paper discusses why previous CGL-based methods can't work well on regression tasks and provides a remedy.
* Molecular Property Prediction: Explaining the effectiveness of contrastive learning for molecular data.

**Theoretical Claims:**

There's no rigorous theoretical claims in this paper. I find some design choices in Section 4 (like selecting identity covariance matrix) are not well supported.

---

> ### Author Rebuttal · Authors · 2025-04-01
>
> We sincerely thank Reviewer iAdk for the useful feedback. Detailed responses are provided below. We kindly ask the reviewer to reconsider their score if the following clarifications resolve the concerns.
>
> ### Q1. Why are gains on GOOD-ZINC larger than on ReactionOOD?
> GOOD-ZINC is large-scale and has diverse OOD testing samples, thus more suitable for causal discovery (which naturally requires more samples than ERM). In contrast, ReactionOOD datasets are smaller and the OOD samples are more specialized. Our method shows larger gains on GOOD-ZINC because our recipe more effectively handles diverse distributions, while baselines struggle with such complexity.
>
> Details can be seen in the response to [Q2 & Claims And Evidence] from Reviewer jru1.
>
> ### W1. Overly narrow scope of causal graph regression
> We respectfully disagree, and humbly refer the reviewer to our response to [Broader Sci. Literature] from Reviewer HaWt.
>
> ### W2. Method motivation
> We humbly refer the reviewer to the response to [Methods] below.
>
> ### [Claims And Evidence] Claim 2 (restated below) lacks support
> > Claim 2: The reason that existing CGL methods don't work well for regression tasks stems from an assumption that confounding subgraphs contain strictly no predictive power.
>
> We respectfully clarify a misunderstanding regarding Claim 2. In our paper, we never claim that existing CGL methods fail in regression tasks due to the assumption that confounding subgraphs contain strictly no predictive power.
>
> Rather, a similar statement serves as part of our **motivation** (instead of claim): prior methods (e.g., CAL [1]) often assume that confounding subgraphs are non-predictive, which does not hold in practice (both in classification and regression settings). We provide empirical evidence supporting this motivation in Section 5.5 (Lines 385–397, Figure 3 left), showing that completely ignoring the predictive signals of confounders can lead to performance degradation.
>
> ### [Methods]
> > - The motivation for using contrastive learning (CL) in regression is unclear.
>
> We respectfully clarify the motivation for using CL. Existing CGL methods (e.g., [1]) proposed to **align the representation** of intervened graphs (constructed by randomly pairing confounding subgraphs with target causal subgraphs [2]) through the **labels of causal subgraphs**, and due to the explicit usage of labels they cannot be directly applied to regression tasks.
>
> From this perspective, CL is a perfect fit for causal graph regression, since the InfoNCE loss in CL aims at "perfect alignment" [3] without using labels, pulling positive sample representations (intervened graphs with the same causal subgraphs in our case) closer while pushing negative pairs (intervened graphs with distinct causal subgraphs) apart, which thus distinguishes causal effects between factual and counterfactual samples. This design enables models to implicitly capture causal signals without label supervision, consistent with recent advances in causal representation learning [4].
>
> > - Is the modeling of $I(C;Y), I(C;G)$ different from those in existing work.
>
> The modeling of $I(C;Y), I(C;G)$ in this paper does differ.
> - Prior works on causal graph learning derive GIB-based objectives under **classification settings**.
> - In contrast, we provide a new formulation of the GIB loss under Gaussian assumptions **specific to graph regression**, allowing principled disentanglement of causal and confounding subgraphs under continuous targets.
>
> ### [Theoretical Claims] Assumption of identity covariance
> We humbly refer the reviewer to our response to W2 of Reviewer HaWt.
>
> ### [Supp. Material] Ablation studies
>
> ![b](https://hackmd.io/_uploads/HkXuO-_ayl.png)
>
> We take your suggestion and re-perform the ablation studies.
>
> Ablation 1 - Effectiveness Analysis: We evaluate the performance of our model variants across four OOD datasets. Each variant removes several proposed modules (the same setting as in Appendix A.5, Lines 597-627).
>
> Due to varying dataset complexities, the contributions of GIB and CI differ across tasks. Nonetheless, the full model consistently achieves the best performance, indicating the two loss functions work synergistically to enhance OOD generalization. (More details can be found in the attached figure.)
>
> Ablation 2 - Parameter Sensitivity Analysis: We humbly refer the reviewer to our response to [Experimental Designs] from Reviewer HaWt.
>
> ### [Reference]
> Thank you for catching this oversight. We will definitely cite the reference when introducing GIB in Sec. 3.2 and acknowledge its influence.
>
> ---
> [1] Causal attention for interpretable and generalizable graph classification. KDD, 2022.
> [2] Debiasing graph neural networks via learning disentangled causal substructure. NeurIPS, 2022.
> [3] Understanding contrastive representation learning through alignment and uniformity on the hypersphere. ICML, 2022.
> [4] Robust causal graph representation learning against confounding effects. AAAI, 2023.

---

> > ### Comment · Reviewer_iAdk · 2025-04-01
> >
> > Thanks for the response and I think it has addressed most of my concerns.

---

> > > ### Author Response · Authors · 2025-04-02
> > >
> > > We sincerely appreciate your valuable suggestions and positive feedback, which will help improve the quality of our manuscript.

---

### Official Review · Reviewer_jru1 · 2025-03-08

**Overall Recommendation:** 3

**Summary:**

This paper investigates causal graph regression (CGR), extending causal graph learning (CGL) techniques, which have been successful in classification tasks, to the more challenging regression setting. The authors introduce a novel approach that adapts causal intervention techniques to regression through the use of contrastive learning. Their method aims to mitigate confounding effects and improve the generalizability of graph neural networks (GNNs) under out-of-distribution (OOD) scenarios. The main claims of the paper include:

1. Contrastive Learning for Causal Graph Regression – The introduction of contrastive learning as a means of handling confounders in graph-level regression tasks.

2. Generalization of Causal Interventions – An adaptation of classification-specific causal intervention techniques to the regression setting.

3. Extensive Empirical Validation on Graph OOD Benchmarks – Demonstrations of effectiveness of the proposed method through experiments on multiple OOD datasets.

##update after rebuttal

**Claims And Evidence:**

The paper provides empirical evidence through experiments on graph OOD benchmarks. The use of contrastive learning to reshape causal intervention in regression is an innovative approach. The results indicate that the proposed method performs well on several datasets, supporting the claim that contrastive learning is effective for CGR. However, the method does not perform well on the Cyloadition and RDB7 datasets. The paper lacks sufficient explanation for these weaker results, which raises concerns about the robustness and generalizability of the approach across diverse datasets.

**Essential References Not Discussed:**

No, all important related works are cited.

**Experimental Designs Or Analyses:**

The experiments are well designed. The analysis on the results on the ReactionOOD dataset can be improved.

**Methods And Evaluation Criteria:**

The proposed method is well-motivated and aligns with the challenges of causal graph regression. The evaluation is conducted on a range of benchmark datasets, which provides a solid empirical foundation

**Other Comments Or Suggestions:**

Typo: Title of Section 5.2, and check line 393 "... loss L_{CI} Conversely ..."

**Other Strengths And Weaknesses:**

N/A

**Questions For Authors:**

Question 1: explain the fundamental difference between graph classification and regression, and why regression is so challenging.

Question 2: hint on why methods such as ERM, CORAL and DANN outperform your method in Table 2 and Table 3.

**Relation To Broader Scientific Literature:**

This work builds on existing research in causal graph learning and graph neural networks, expanding causal intervention techniques beyond classification to regression tasks.

**Theoretical Claims:**

The variational bound for the GIB objective seems fine to me.

---

> ### Author Rebuttal · Authors · 2025-04-01
>
> We sincerely thank Reviewer jru1 for the thoughtful and encouraging feedback. We appreciate your recognition of our method’s motivation, design, and empirical support. Below we address each of your comments in detail.
>
> ### Q1. Explain the fundamental difference between graph classification and regression, and why regression is so challenging.
>
> The core difference lies in the adoptation of causal discovery, as noted in Lines 057–060, where we state that existing methods "rely on **discrete label information** and cannot be adapted to regression."
>
> In regression, the absence of discrete labels naturally makes confounder separation harder. For example, in molecular property prediction, continuous outcomes (e.g., solubility values) often arise from overlapping substructures, complicating causal identification.
>
> ### [Q2 & Claims And Evidence] Performance Gaps on ReactionOOD Datasets
>
> We appreciate this observation. The relatively weaker performance is attributable to known challenges rather than methodological flaws:
> * Our method stems from **more general and weaker assumptions**, which naturally implies larger function space and higher requirements on sample size:
> Unlike prior methods such as IRM or DIR, our method do not assume that spurious features are non-predictive. This allows us to model a broader function class under weaker assumptions, making our method applicable to a wider range of real-world distribution shifts. The trade-off is increased optimization difficulty under limited data or high spurious signal, but this reflects a principled design choice, not a methodological weakness.
> * No method dominates across all OOD benchmarks:
> As shown in OOD-GNN [1], it is a regular phenomenon in generalization tasks that no approach consistently performs best on every dataset due to varying distribution shifts and inductive biases. Despite this, our method is among the most stable across diverse settings and achieves best results on ZINC (Table 1), the largest and most complex dataset.
> * Spurious correlations affect ID performance:
> Non-causal baseline methods (e.g., ERM, CORAL, and DANN) often perform better under ID or mild OOD due to exploiting spurious but predictive features. To handle OOD settings, our method instead recognizes those spurious features and removes them for better generalization, which may slightly reduce ID performance but improves robustness.
>     > For example, in Cycloaddition-ID（Total Atom Number, concept), CORAL achieves 4.10 vs. ours 5.74, but under OOD, CORAL drops to 5.74 while ours remains stable at 5.53 (the smaller the better).
> * Causal and contrastive learning require more data:
> Causal inference is statistically harder than correlation-based methods and generally requires more data to reduce variance [2]. Contrastive learning also benefits from large batch sizes and diverse representations. This explains why our method performs best on ZINC. On smaller datasets (e.g., RDB7), all causal methods degrade; however, our method remains stable and consistently outperforms CIGA (the best method of the listed causal intervention baselines) across all RDB7 settings.
>
>
> We will modify the corresponding experimental analysis.
>
>
> ### [Typo] Correct typo in Section 5.2 title and line 393.
>
> We appreciate your attention to detail. We will accordingly revise the manuscript to reflect your suggestion.
>
> ---
> [1] Tajwar F et al. No true state-of-the-art? OOD detection methods are inconsistent across datasets. arXiv, 2021.
> [2]  Guo R et al. A survey of learning causality with data: Problems and methods. ACM Comput. Surv., 2020.

---

### Official Review · Reviewer_HaWt · 2025-03-12

**Overall Recommendation:** 2

**Summary:**

This paper proposes an improved causal graph regression method by an enhanced graph information bottleneck loss function and a contrastive learning loss from generated counterfactual graphs. Experiments on OOD datasets confirm its generality.

**Claims And Evidence:**

Yes

**Essential References Not Discussed:**

I do not know any related works that are essential to this paper.

**Experimental Designs Or Analyses:**

Yes, I checked the experiments. The results verify the effectiveness of the proposed method. The only issue is that there is no discussion on the effect of the hyperparameters $\alpha$ and $\beta$, which seem to be important hyperparameters in the method.

**Methods And Evaluation Criteria:**

Yes

**Other Comments Or Suggestions:**

"5.2. baselines and Setup " -> "5.2. Baselines and Setup"

**Other Strengths And Weaknesses:**

- The paper is well-written and easy to follow.
- The experimental results are good.

1. Could you please explain how to obtain the causal subgraph $C$ from a graph $G$, i.e., how to obtain the mask matrices $M_{edge}$ and $M_{node}$? Even though the concrete approach might be discussed in previous work, I think it would be better to have a brief introduction in the background to make the paper more consistent.
2. There are several assumptions in Section 4.2. However, it remains unknown whether these assumptions are reasonable in practice.
3. The OOD generality is claimed as a core contribution of the proposed approach. However, there is no obvious superiority of the OOD performance compared with the ID performance.

**Questions For Authors:**

See above.

**Relation To Broader Scientific Literature:**

Causal graph regression seems to be a relatively neglected domain. The authors also do not provide any important applications of the causal graph regression task. Therefore, I think the key contributions of the paper may not have a large influence to the broader scientific literature.

**Theoretical Claims:**

There are no formal theoretical claims in this paper.

---

> ### Author Rebuttal · Authors · 2025-04-01
>
> We sincerely thank Reviewer HaWt for the thorough and constructive feedback. We have carefully addressed the raised concerns and suggestions. We kindly ask the reviewer to reconsider their score if the following points resolve the reservations with our work, or to provide more pointers for us to address.
>
> ### W1. How to obtain the causal subgraph $C$ from graph $G$, i.e., how to obtain the mask matrices $M_{edge}$ and $M_{node}$.
>
> We appreciate the reviewer's question. The construction of the causal subgraph is introduced in the main text (Eq. (1)), and the details of generating the mask matrices are in Appendix B (Lines 647–692).
>
> In the revision, we will briefly summarize the mask generation process in the main text.
>
> ### W2. The assumptions in Section 4.2 remain unverified. Are they reasonable?
>
> Thank you for raising this point. We go through and clarify each assumption below:
> * Assumption 1 (Gaussian assumption of $p(C \mid G)$ and $q(C)$):
> We follow the variational information bottleneck literature [4], modeling both as multivariate Gaussians to enable tractable KL estimation. This is widely adopted in similar works [5,6].
> * Assumption 2 (Covariance $\Sigma_\phi(G) = I$):
> Setting the covariance to identity simplifies optimization and is theoretically justified in [4, Appendix A], which shows that any full-rank covariance can be whitened without loss of generality.
> * Assumption 3 (Gaussian posterior):
> For $P(Y \mid H_c)=\mathcal{N}(Y; \mu_{(c)}, \sigma^2_{(c)})$, we use a fixed variance for stability, similar to Phase I assumption in section 1 of [7]. This also allows a closed-form expression for mutual information estimation (e.g., [6], Sec.2.1, Proposition 1).
>
> We will clarify these assumptions more explicitly and emphasize their empirical validity (the strong OOD generalization results in Section 5) in the revision.
>
> ### W3. There is no obvious superiority of the OOD performance compared with the ID performance.
>
> We respectfully clarify that the goal of CGL methods might be misunderstood; we do not aim for OOD performance to surpass ID performance, which is unrealistic.
>
> We focus our attention on **the robustness under OOD settings**, demonstrating strong generalization ablity of our method. For more analysis of OOD performance, we humbly refer the reviewer to the response to Q2 from Reviewer jru1.
>
> ### [Experimental Designs] The effect of the hyperparameters $\alpha$ and $\beta$
> ![a](https://hackmd.io/_uploads/H1wfF-upyx.png)
>
> We thank the reviewer for this suggestion. Some discussion related to this point has already been included in Appendix A.5. To better illustrate, we rerun the experiments and will update the sensitivity analysis for $\alpha$, $\beta$, and $\lambda$. (More details can be find in the attached figure.)
>
> The results show that the choice of $\alpha$ has no clear effect, while the choices of $\beta$ and $\lambda$ significantly impact OOD performance.
>
> ### [Broader Sci. Literature] Causal graph regression (CGR) seems to be a relatively neglected domain, and the authors also do not provide any important applications of CGR
>
> We appreciate this review and will revise the introduction to better highlight the motivating applications and clarify the contribution as **improving generalization in graph-based prediction tasks through causal mechanisms**.
>
> Furthermore, we beg to clarify
> - CGR is not a niche topic but a core component of the broader field of **causal graph learning** (CGL). As noted in Section 1 (lines 36-46), CGL has emerged as a key paradigm for building robust models. Our work leverages causal interventions to improve OOD generalization in graph regression, which is essential for many real-world scenarios with distributional shifts.
> - We have discussed the application of CGR in Section 1 (lines 48-51). Moreover, CGR is relevant to tasks such as molecular property prediction [1], traffic flow forecasting [2], and credit scoring [3].
>
> ---
> [1] Rollins Z A. et al. MolPROP: Molecular Property prediction with multimodal language and graph fusion. J. Cheminform., 2024.
> [2] Li G. et al. Multistep traffic forecasting by dynamic graph convolution: Interpretations of real-time spatial correlations. Transp. Res. Part C Emerg. Technol., 2021.
> [3] Ma F. et al. Utilizing Reinforcement Learning and Causal Graph Networks to Address the Intricate Dynamics in Financial Risk Prediction. Int. J. Inf. Technol. Syst. Approach, 2024.
> [4] Chechik G. et al. Information bottleneck for Gaussian variables. Adv. Neural Inf. Process. Syst., 2003.
> [5] Kingma D P, Welling M. Auto-encoding variational bayes[EB/OL].(2013-12-20)
> [6] Yu S. et al. Cauchy-Schwarz Divergence Information Bottleneck for Regression. arXiv, 2024.
> [7] Nix D. et al. Learning local error bars for nonlinear regression. Adv. Neural Inf. Process. Syst., 1994.

---

> > ### Comment · Reviewer_HaWt · 2025-04-02
> >
> > I appreciate the detailed response. My concerns on assumptions, OOD performance, and hyperparameter settings are addressed.
> >
> > > The construction of the causal subgraph is introduced in the main text (Eq. (1)), and the details of generating the mask matrices are in Appendix B (Lines 647–692).
> >
> > Thank you for pointing out Appendix B. I am still confused on how MLP_edge and MLP_node are trained in Eq. (19) - (20), and which is the GNN-based encoder $f$. In the description of Sec. 3.1, it seems that $C$ and $S$ can be obtained directly and deterministically from $G$; while in Appendix B, it seems that we need additional neural networks $f$ and MLPs, and it is unknown where these neural networks come from. And I think Appendix B is more like a detailed explanation of the proposed framework instead of Sec. 3.1.
> >
> > > CGR is not a niche topic but a core component of the broader field of causal graph learning (CGL). As noted in Section 1 (lines 36-46), CGL has emerged as a key paradigm for building robust models. Our work leverages causal interventions to improve OOD generalization in graph regression, which is essential for many real-world scenarios with distributional shifts.
> > We have discussed the application of CGR in Section 1 (lines 48-51). Moreover, CGR is relevant to tasks such as molecular property prediction [1], traffic flow forecasting [2], and credit scoring [3].
> >
> > Yes, I agree that CGL is crucial to being a key paradigm for building robust models. The key problem is whether CGR is significant, as it is also mentioned that "previous CGL studies focus on classification settings." As this paper seems to be one of the first papers emphasizing the regression task, I think it would be helpful to provide some stronger supports for the importance of the CGR task. Therefore, I suggest mentioning these applications [1-3] in the revised version.

---

> > > ### Author Response · Authors · 2025-04-04
> > >
> > > We sincerely thank the reviewer for the constructive feedback and thoughtful questions. We are particularly grateful that the reviewer acknowledged our efforts; as noted in your response, the concerns on assumptions, OOD performance, and hyperparameter settings have been addressed, and we appreciate your positive recognition.
> > >
> > > ### Q1.
> > >
> > > > Thank you for pointing out Appendix B. I am still confused on how MLP_edge and MLP_node are trained in Eq. (19) - (20), and which is the GNN-based encoder f. In the description of Sec. 3.1, it seems that C and S can be obtained directly and deterministically from G; while in Appendix B, it seems that we need additional neural networks f and MLPs, and it is unknown where these neural networks come from. And I think Appendix B is more like a detailed explanation of the proposed framework instead of Sec. 3.1.
> > >
> > >
> > > Thank you for pointing out this ambiguity. Below we manage to clarify the confusion regarding **(1)** how to get C and S from G; **(2)** how MLP_edge/node are trained.
> > >
> > >
> > > (1) how to get C and S from G
> > >
> > > * **The function $f(\cdot)$ always** refers to a GNN encoder along this paper, which takes the input graph $G=(A, X)$ and produces a graph-level embedding $H_g =f(A, X)$.
> > > * Based on $H_g$, **two MLPs** (MLP_edge and MLP_node) are used to generate soft attention masks $M_{node}$ and $M_{edge}$, which assign importance scores to nodes and edges. We then follow Eq. (1) to compute $C = (M_{edge} \odot A, M_{node} \odot X)$. Therefore, the subgraph $C$ cannot be obtained directly through $G$ (similarly for $S$); the masks $M_{edge}, M_{node}$ are returned by MLP_edge, MLP_node in Eqs. (19) - (20).
> > >
> > > (2) how MLP_edge/node are trained
> > >
> > > * After obtaining $C$, we continue the forward pass, encoding $C$ ($S$ doesn't explicitly appear in the loss) and passing the representation of $C$ to the final loss (line 246) $L_c(G,C,Y)=-I(C;Y)+\alpha I(C;G)$. Since the loss involves the representation of $C$, which is computed via the two MLPs, the parameters of **MLP_edge and MLP_node are thus trained end-to-end via backpropagation**.
> > >
> > > For your reference, we provide a road map for the related contents in our submission.
> > > - The specific procedure for **generating $C$ and $S$** is summarized in Sec. 3.1
> > > - The technical details of the **whole prediction process** above are provided in Appendix B (Eqs. 18-20).
> > > - Our full pipeline, including **the proposed loss and overall framework** built upon this foundation, is illustrated in Figure 2 and described in Sec. 4.1.
> > >
> > > We will revise Sec. 3.1 (on how $C$ and $S$ are generated) to clearly state that the masks are learnable and produced by trainable components. Thank you again for pointing this out.
> > >
> > > ### Q2
> > >
> > > > Yes, I agree that CGL is crucial to being a key paradigm for building robust models. The key problem is whether CGR is significant, as it is also mentioned that "previous CGL studies focus on classification settings." As this paper seems to be one of the first papers emphasizing the regression task, I think it would be helpful to provide some stronger supports for the importance of the CGR task. Therefore, I suggest mentioning these applications [1-3] in the revised version.
> > >
> > > Thank you for your valuable suggestion. We will highlight its significance more clearly and include the mentioned applications in the revised version.
> > >
> > > Moreover, we would like to gently emphasize that CGR is an important yet underexplored task, partially due to its inherent difficulty in modeling. Some molecular property prediction tasks are natively regression problems, while subsequently reformulated as classification problems to facilitate solution (e.g., Tox21 [8]). We will also add this discussion to the next revision.
> > >
> > > ---
> > > [8] Huang R et al. Tox21Challenge to build predictive models of nuclear receptor and stress response pathways as mediated by exposure to environmental chemicals and drugs. Front. Environ. Sci., 2016.

---

### Decision · Program_Chairs · 2025-05-01

**Decision:**

Accept (poster)

**Comment:**

The work deals with the problem of regression using a causal graph where the most important subgraph from a given input graph can be learned for the given task at hand using a graph neural network. As claimed by the authors, this task, termed "causal graph learning," is a relatively new problem, and the major focus has been on classification tasks. This work extends the literature towards the regression task and adapt causal intervention techniques to regression through the use of contrastive learning.

The ratings of the paper lean towards the positive, with the issue that no reviewer is willing to champion the paper. The AC read the paper and believes that the paper proposes a solid contribution to a niche area of research and can be an interesting paper to discuss among the causal and GNN community in the conference. I thus recommend acceptance.